# In Utero Dexamethasone Exposure Exacerbates Hepatic Steatosis in Rats That Consume Fructose During Adulthood

**DOI:** 10.3390/nu11092114

**Published:** 2019-09-05

**Authors:** Tanyara B. Payolla, Caio J. Teixeira, Fabio T. Sato, Gilson M. Murata, Gizela A. Zonta, Frhancielly S. Sodré, Carolina V. Campos, Filiphe N. Mesquita, Gabriel F. Anhê, Silvana Bordin

**Affiliations:** 1Department of Physiology and Biophysics, Institute of Biomedical Sciences, University of Sao Paulo, Sao Paulo 05508-000, Brazil; 2Department of Pharmacology, Faculty of Medical Sciences, State University of Campinas, Sao Paulo 13083-887, Brazil

**Keywords:** fructose, hepatic steatosis, glucocorticoids, pregnancy, DOHaD

## Abstract

Distinct environmental insults might interact with fructose consumption and contribute to the development of metabolic disorders. To address whether in utero glucocorticoid exposure and fructose intake modulate metabolic responses, adult female Wistar rats were exposed to dexamethasone (DEX) during pregnancy, and the offspring were administered fructose at a later time. Briefly, dams received DEX during the third period of pregnancy, while control dams remained untreated. Offspring born to control and DEX-treated mothers were defined as CTL-off and DEX-off, respectively, while untreated animals were designated CTL-off-CTL and DEX-off-CTL. CLT-off and DEX-off treated with 10% fructose in the drinking water for 8 weeks are referred to as CTL-off-FRU and DEX-off-FRU. We found that fructose promoted glucose intolerance and whole-body gluconeogenesis in both CTL-off-FRU and DEX-off-FRU animals. On the other hand, hepatic lipid accumulation was significantly stimulated in DEX-off-FRU rats when compared to the CTL-off-FRU group. The DEX-off-FRU group also displayed impaired very-low-density lipoprotein (VLDL) production and reduced hepatic expression of *apoB*, *mttp*, and *sec22b*. DEX-off-FRU has lower hepatic levels of autophagy markers. Taken together, our results support the unprecedented notion that in utero glucocorticoid exposure exacerbates hepatic steatosis caused by fructose consumption later in life.

## 1. Introduction

Since the initial description of hypertriglyceridemia and the impairment of insulin-mediated glucose clearance induced by a fructose-enriched diet in rats [1], several studies with animal models have described consistent changes in energy metabolism caused by consuming excess fructose in the drinking water. In fact, this hypertriglyceridemia has been attributed to either increased hepatic production or slower clearance of very-low-density lipoprotein (VLDL) [2,3,4,5], while the observed glucose intolerance was ascribed to insulin resistance in the skeletal muscle or increased hepatic glucose production [6,7,8].

Despite the numerous animal studies on the topic, the cause/effect relationship between fructose consumption and increased rates of obesity and cardiometabolic disorders in Western countries remains controversial [9]. The observed weight gain associated with the consumption of sugar-sweetened beverages appears to be marginal if other variables such as smoking status, age, or physical activity are taken into account [10]. Additionally, randomized controlled trials have reported variable fructose-related impacts on triglyceride and LDL content [11,12,13]; thus, it is unlikely that fructose consumption alone could be responsible for the increased incidence of metabolic disorders in humans [9].

In fact, it is reasonable to assume that multiple environmental factors to which humans are exposed interact with components of the diet and facilitate the development of dyslipidemia and insulin resistance. In accordance with the early studies that gave rise to the “developmental origins of health and disease” (DOHaD) concept [14], such environmental exposure can occur in utero.

Although the DOHaD concept has initially focused on the nutritional status during pregnancy, fetal overexposure to glucocorticoids—a common feature of maternal undernutrition, maternal stress, and poor fetal blood supply—has also been shown to play a central role in the metabolic programming of the offspring [15]. For example, previous studies demonstrated that in utero dexamethasone (DEX) exposure favors hepatic steatosis in the adult offspring subjected to either food deprivation or excess fat intake [16,17,18]. It is also worth noting that exposure to antenatal glucocorticoids is far from being an exclusively experimental condition, since its use is a common medical approach for increasing the pre-term survival rate [19].

With regards to the modulation of fructose metabolism, glucocorticoids have been shown to reduce fructose conversion into fatty acids in adipose tissue or to stimulate its conversion into glycogen in the liver [20,21]. It is also known that early glucocorticoid exposure increases intestinal α-glucosidase activity, thus boosting sucrose digestion and fructose absorption [22,23,24].

The present study sought to investigate the relationship between in utero glucocorticoid exposure and fructose metabolism by evaluating the metabolic consequences of eight weeks of liquid fructose supplementation in the adult male offspring of DEX-treated pregnant rats.

## 2. Materials and Methods

### 2.1. Experimental Design and Diet

Eight-week-old nulliparous Wistar rats were housed in a temperature-controlled room (22 ± 2 °C) with the lights on from 7:00 a.m. to 7:00 p.m.; standard chow (NUVILAB^®^ Cr-1, Nuvital, Colombo, PR—Brazil) (Table 1) and water were available ad libitum. After a two-week acclimation period, the female rats were housed with male rats for 5 days (two females with one male per cage). The concomitant presence of spermatozoa and estrous cells in a vaginal lavage indicated Day 0 of gestation. At the 14th day of pregnancy, rats were randomized into two groups, with one group receiving untreated drinking water and the other receiving 0.2 mg/kg/day of water-soluble DEX diluted in the drinking water for five days.

After birth, the offspring born to CTL mothers (CTL-off) and DEX-treated mothers (DEX-off) were adjusted to consist of six pups per lactating mother and were allowed to lactate until postnatal Day 21. After weaning, the CTL-off and DEX-off were fed a standard chow diet and provided with water ad libitum until the 8th week of age (first period of offspring life). At the 8th week of life, a portion of the CTL-off and DEX-off animals continued to have free access to untreated drinking water (CTL-off-CTL and DEX-off-CTL), while the drinking water of the remaining animals contained 10% fructose (w/v) (CTL-off-FRU and DEX-off-FRU). The 10% fructose treatment lasted until the 16th week of age (second period of offspring life).

All experiments were conducted with 12 h fasting by removing the chow at 7:00 p.m. and conducting experiments in the following morning. Tap water was offered ad libitum.

All studies were performed according to the guidelines of the Brazilian College for Animal Experimentation (COBEA) and approved by the Ethics Committee on Animal Use at the Institute of Biomedical Sciences, University of Sao Paulo, Brazil.

### 2.2. Analysis of Blood Parameters

Glucose, triglycerides, and cholesterol determinations were performed using commercially available kits (Labtest Diagnóstica SA, Lagoa Santa, MG, Brazil).

### 2.3. Analysis of Tissue Parameters

Liver samples were quickly removed, washed with ice-cold PBS, frozen with liquid nitrogen, and ground in a mortar. Powdered samples were maintained in a deep freezer at −80 °C until use.

The glycogen and lactate content was measured as previously described [18,25]. Lipids were extracted from liver samples using a previously described method [26]. Briefly, powdered samples were homogenized using a rotor-stator in a 4 mL solution of CHCl_3_ and methanol (2:1, v/v). The homogenate was subjected to extraction for at least 16 h at 4 °C with gentle homogenization in a closed glass tube. After this, a 0.6% solution of NaCl was added to the extract and centrifuged at 2000× *g* for 20 min. The organic layer was collected and dried in an Eppendorf Vacuum Concentrator Plus (Eppendorf, Hamburg, Germany). The lipids were solubilized in 200 µL of isopropanol and quantified using commercially available kits.

The activities of pyruvate kinase (PK, EC 2.7.1.40), glucose 6-phosphate dehydrogenase (G6PDH: EC 1.1.1.49), lactate dehydrogenase (LDH; EC 1.1.1.27), and phosphoenolpyruvate carboxykinase (PEPCK; EC 4.1.1.32) were measured using spectrophotometric assays, following standard methods described elsewhere [27,28,29]. Citrate synthase (CS; EC 4.1.3.7) and carnitine palmitoyltransferase (CPT1; EC 2.3.1.21) activities were assayed at 412 nm as previously described [30,31].

### 2.4. Quantitative Real-Time Polymerase Chain Reaction (PCR)

The analysis of mRNA and its expression was performed as previously described [25]. Briefly, the total RNA was extracted using TRIzol reagent from approximately 100 mg of tissue and subsequently used for reverse transcription with random primers. The primer sequences and accession numbers were as follows: apolipoprotein B (*apoB*; NM_019287) sense 5′-CTGCGGTGGCAGAAATAACG-3′ and antisense 5′-CCTTGAGCAAACCTTAGGTAGGG-3′; acetyl-CoA carboxylase alpha (*acca*; NM_022193) sense 5′-TGCTTATATTGTGGATGGCTTG-3′ and antisense 5′-TTCTACTGTCCCTTCTGGTTCC-3′; fatty acid synthase (*fasn*; NM_017332) sense 5′-TGGTGAAGCCCAGAGGGATC-3′ and antisense 5′-CACTTCCACACCCATGAGCG-3′; microsomal triglyceride transfer protein (*mttp*; NM_001107727) sense 5′-TATGACCGTTTCTCCAAGAGTGG-3′ and antisense 5′-TCAAGGTTCTCCTCTCCCTCATC-3′; SEC22 vesicle trafficking protein homolog B (*sec22b*; NM_001025686) sense 5′-CGTGCTCGGAGAAATCTCGG-3′ and antisense 5′-AACACGGCTACTGCTGCAAGC-3′; and ribosomal protein L37a (*rpl37a*; X14069) sense 5′-CAAGAAGGTCGGGATCGTCG-3′ and antisense 5′- ACCAGGCAAGTCTCAGGAGGTG-3′. Values of mRNA expression were normalized using the internal control gene *Rpl37a* [18]. Fold changes were calculated by the 2^−ΔΔCT^ method.

### 2.5. Immunoblotting

Fragments of liver (approximately 100 mg) were removed and processed for Western blotting as previously described [25]. The primary antibodies used were anti-Beclin 1 (BECN1) (cat. # sc-48341) from St. Cruz Biotechnology (Dallas, TX, USA), anti-microtubule associated protein 1 light chain 3 alpha (LC3) (cat. # ab48394), and anti-sequestosome 1 (SQSTM1/p62) (cat. # ab91526) from Abcam (Cambridge, UK). Secondary antibodies conjugated to horseradish peroxidase (Bio-Rad Laboratories, Hercules, CA, USA) were employed for chemiluminescent detection of the protein bands, and an Amersham Imager 600 (GE Healthcare Waukesha, WI, USA) was employed for visualization. Band intensities were quantified by optical densitometry using ImageJ software (http://imagej.nih.gov/ij). Nitrocellulose membranes were routinely stained with Ponceau S and scanned before being incubated with the antibodies. The digital images of the Ponceau S-stained membranes were subjected to optical densitometry and used to normalize the chemiluminescent signal of the bands.

### 2.6. Intraperitoneal Glucose Tolerance Test (ip-GTT)

At 16 weeks of age, the male offspring were fasted and administered glucose by intraperitoneal (IP) injection (2 g of D-glucose/kg body weight). Blood glucose measurements were performed with blood samples collected from the tail at 0, 10, 15, 30, 60, and 120 min after the injection. Glucose tolerance was based on the area under the curve (AUC) and was measured from above each individual baseline (basal glycemia) for glycemia vs. time.

### 2.7. Intraperitoneal Pyruvate Tolerance Test (ip-PTT)

At 16 weeks of age, the male offspring were fasted and administered pyruvate by IP injection (2 g of a sodium pyruvate/kg body weight). Blood samples were collected from the tail at 0, 10, 15, 30, 60, and 120 min after the injection. To estimate pyruvate tolerance, the AUC of glycemia vs. time was measured from above each individual baseline (basal glycemia).

### 2.8. VLDL Production Assay

The Lipoprotein Lipase (LPL) inhibitor tyloxapol (Triton WR-1339; Sigma Aldrich, St Louis, MO, USA) was dissolved in isotonic saline 0.9% (v/v) under slight agitation. Then, after a 12 h fast, rats were weighed and received an IP injection of tyloxapol (500 mg/kg). Blood samples were collected form the tip of the tail at 1, 2, 3, and 6 h after the injection, and the triglyceride content was immediately measured using a commercially available diagnostic kit (Labtest, Santa Lagoa, Brazil). The slopes of the curves, obtained from the linear regressions, were used to estimate the VLDL production rate.

### 2.9. Statistical Analysis

All results are presented as the means ± standard error (SE). Comparisons were performed using two-way ANOVA, followed by a Tukey’s multiple comparison test. The two factors considered for the two-way ANOVA were in utero DEX exposure and fructose consumption during adulthood. When making comparisons between two groups the unpaired Student’s *t*-test was utilized. Statistical analyses were conducted using GraphPad Prism software version 8.1.1 (GraphPad Software, Inc., San Diego, CA, USA). Results with *p* values that were less than 0.05 were considered significant.

## 3. Results

### 3.1. Effect of In Utero DEX Exposure on Body Weight Gain and Fructose Intake

A summary of the birth weight, body weight gain, and caloric intake for each group is presented in Table 2. We found that in utero DEX exposure significantly reduced the birth weight of the male offspring by 52% when compared to the offspring born to CTL mothers (*p* < 0.0001). Additionally, the body weights of rats born to DEX-treated mothers remained reduced by 47% and 11% at the 3rd and 8th weeks of age, respectively, when compared to the offspring born to CTL mothers (*p* = 0.0002 and *p* = 0.01, respectively). However, the body weight gains of both groups were observed to be similar during the first period of life (i.e., weaning to 8 weeks of age). Body weight gains throughout the second period of offspring life and body weights at the 16th week of life were lower in rats born to DEX-treated mothers (respectively, *p* = 0.003 and *p* < 0.0001). Subgroup analysis revealed that body weight gain throughout the second period of offspring life and body weights at the 16th week of life were higher in CTL-off-FRU than in DEX-off-FRU (respectively, 55% and 16% higher; *p* = 0.02 and *p* = 0.04).

Furthermore, both CLT-off-FRU and DEX-off-FRU displayed reduced amounts of solid caloric intake during the second period (*p* < 0.001). In contrast, the CTL-off-FRU group consumed 30% more liquid calories than did DEX-off-FRU rats (*p* < 0.001). Overall, the total caloric intake during the second period of life was higher in offspring that were administered liquid fructose (*p* < 0.05).

### 3.2. Effect of Fructose Consumption on Glucose Tolerance and Gluconeogenesis in Offspring Born to CTL and DEX-Treated Mothers

To determine whether glucose metabolism was differentially modulated by fructose intake in offspring born to DEX-treated mothers, fructose was included in the drinking water for eight weeks, and GTTs and PTTs were performed. As shown in Figure 1A, animals that consumed fructose, regardless of DEX exposure, exhibited increased AUCs in the GTTs when compared to animals that did not consume fructose (*p* < 0.0001). It should be pointed out that the AUCs of the GTTs increased by 196% and 255% in the CTL-off-FRU and DEX-off-FRU groups, respectively, when compared to the age-matched CTL-off-CTL (*p* < 0.0001). Additionally, the age-matched DEX-off-CTL group also did not present any signs of glucose intolerance, appearing almost indistinguishable from CTL-off-CTL.

In order to assess the rate of whole-body gluconeogenesis, glucose production was monitored after a pyruvate challenge (Figure 1B). Similar to what was observed with the GTT, fructose consumption increased the AUCs after the pyruvate injections when compared to animals that did not receive fructose (*p* < 0.0001). Additionally, these AUC values of the CTL-off-FRU and DEX-off-FRU groups were increased by 131% and 151%, respectively, when compared to that of CTL-off-CTL (*p* < 0.0001). There was also no evidence for age-matched DEX-off-CTL undergoing changes in whole-body gluconeogenesis when compared to CTL-off-CTL.

### 3.3. Effect of Fructose Consumption on Adiposity and Hepatic Triglyceride Content in Offspring Born to CTL and DEX-Treated Mothers

As shown in Figure 2A–C, changes in adiposity were evaluated in CTL-off-CLT, DEX-off-CLT, CLT-off-FRU, and DEX-off-FRU rats. The relative weights of mesenteric fat pads were similarly increased in DEX-off-CTL, CTL-off-FRU, and DEX-off-FRU animals, with masses that were 220% (*p* < 0.01), 296% (*p* < 0.001), and 169% (*p* < 0.05) greater than those measured in CTL-off-CTL animals, respectively (Figure 2A). In contrast, the relative weights of epididymal fat pads were increased in DEX-off-CTL and CTL-off-FRU animals by 34% (*p* < 0.01) and 37% (*p* < 0.001), respectively, but no such change was noted in the DEX-off-FRU samples (*p* = 0.0003 for interaction) (Figure 2B). The relative weights of retroperitoneal fat pads were increased by 36% in CTL-off-FRU animals when compared to the CTL-off-CTL group (*p* < 0.001), but no increase was observed in the DEX-off-FRU (*p* = 0.0014 for interaction) (Figure 2C). The hepatic triglycerides content was increased by treatment with liquid fructose during adult life when taking into account animals born to both CTL and DEX mothers (27% higher than animals not treated with fructose; *p* < 0.0001). Besides this expected result of fructose intake, our sub-group analysis revealed that the hepatic triglycerides content in DEX-off-FRU was 51% higher than that in CTL-off-FRU (*p* < 0.0001) (Figure 2D).

### 3.4. Changes in Hepatic Metabolism Cannot Account for the Augmented Liver Triglyceride Content Induced by Fructose Treatment in Offspring Born to DEX-Treated Mothers

Due to the fact that intermediary fructose metabolites in the liver can enter glycolysis and generate glucose, glycogen, or fatty acids, we evaluated the activity of key enzymes belonging to these pathways in order to understand the exacerbation of liver triglycerides content seen in the rats belonging to the DEX-off-FRU group.

Our data demonstrate that the hepatic glycogen content was equally increased by fructose treatment in CTL-off-FRU and DEX-off-FRU (approximately 80% higher than in CTL-off-CTL; *p* < 0.0001) (Figure 3A). Similarly, the activity of PEPCK, a limiting step to shunt glycolysis-originated oxalacetate to gluconeogenesis, was increased by treatment with liquid fructose during adult life when taking into account animals born to both CTL and DEX mothers (18% higher than animals not treated with fructose; *p* < 0.01). Moreover, the PEPCK activity levels of both CTL-off-FRU and DEX-off-FRU were similar between each other and higher than that of CTL-off-CTL (respectively, 14% and 13% higher; *p* = 0.002 and *p* = 0.003) (Figure 3B).

Additionally, we evaluated the consumption of hepatic glycolysis-generated fructose metabolites by measuring PK activity. There was a trend for the fructose-consuming experimental groups to display increased PK activity (*p* = 0.0065), but only the results from CTL-off-FRU rats (72% increase) were found to be significant when compared to CTL-off-CTL (*p* = 0.02) (Figure 3C).

We evaluated lactate production by assessing hepatic LDH activity and lactate content. Our data show that fructose treatment increased both LDH activity (Figure 3D) and lactate content (Figure 3E), as compared to the CLT-off-CLT group (*p* = 0.0061 and *p* = 0.0014, respectively). However, when compared to the CTL-off-CLT group, only CTL-off-FRU rats presented significantly altered activity (39% increase, *p* = 0.002) and content (48% increase, *p* = 0.01) (Figure 3D,E).

To further investigate the role of fructose metabolites, we also evaluated fatty acid synthesis by conducting enzymatic assays and qPCR. A significant increase in CS activity was observed in CTL-off-FRU rats when compared to the CTL-off-CTL (38% increase, *p* = 0.02) and DEX-off-FRU (43% increase, *p* = 0.01) groups (Figure 4A). Additionally, G6PDH activity, a limiting step of the pentose phosphate pathway (PPP), was found to be increased in both groups receiving fructose, with values that were approximately 80% higher than for those not treated with fructose (*p* < 0.0001). Nevertheless, G6PDH activity in CTL-off-FRU was similar to that in DEX-off-FRU (Figure 4B). It was also found that CPT1 activity, a pivotal enzyme for fatty acid oxidation, was not perturbed in any of the experimental groups used in the present study (Figure 4C).

With regards to de novo lipogenesis (DNLG), we monitored *fasn* and *acca* in each of the experimental groups. It was determined that exposure to fructose upregulated *fasn* and *acca* expression levels by 93% (*p* = 0.04) and 81% (*p* < 0.0003), respectively. Subsequently, subgroup analyses revealed that these genes were expressed at similar levels in both CTL-off-FRU and DEX-off-FRU rats (Figure 4D).

### 3.5. Fructose-Induced Accumulation of Hepatic Triglycerides is Accompanied by Impaired VLDL Production in the Livers of Rats Born to DEX-Treated Mothers

We next evaluated several parameters associated to VLDL assembly and secretion since none of the changes in enzymatic activity could account for the exacerbated hepatic triglyceride accumulation observed in the DEX-off-FRU group. As shown in Figure 5, cholesterol (Figure 5A) and triglyceride levels (Figure 5B) were increased by 21% and 98%, respectively, following fructose treatment when compared to animals not treated with fructose (*p* = 0.0003 and *p* < 0.0001, respectively). Notably, this increase was blunted in the DEX-off-FRU group, which had triglyceride levels that were reduced by 47% as compared to DEX-off-FRU rats (*p* = 0.01) (Figure 5B).

The expression levels of *sec22b*, *mttp*, and *apob* were evaluated due to their pivotal role in VLDL assembly and secretion [25]. The expression of all these genes was increased in the CLT-off-FRU group (*p* < 0.01). In fact, the expression levels of *sec22b* and *mttp* were increased by 92% and 147%, respectively, when compared to the levels detected in DEX-off-FRU rats (*p* = 0.02 and *p* = 0.01, respectively). Additionally, subgroup analyses also revealed that the increased *apob* expression (205%) in CTL-off-FRU animals was significantly different from that in the CTL-off-CTL group (*p* = 0.005) (Figure 5C).

In order to measure hepatic VLDL production, serum triglyceride concentrations were also evaluated after a challenge with tyloxapol. As shown in Figure 5D, the blood triglyceride concentration 6 h after tyloxapol injection was 201% higher in CTL-off-FRU than in DEX-off-FRU (*p* < 0.0001). The VLDL production rate was also found to be increased in the CTL-off-FRU group when compared to DEX-off-FRU (*p* = 0.03) (Figure 5D).

### 3.6. Effect of Fructose on the Expression of Autophagy Markers in the Livers of the Offspring Born to DEX-Treated Mothers

Autophagy flux has been implicated in the regulation of cellular lipid stores and hepatic steatosis. Thus, we evaluated three markers—Beclin1, p62, and LC3B—of autophagy flux and attempted to elucidate the underlying mechanisms involved with the increased triglycerides stores and reduced VLDL secretion observed in the DEX-off-FRU animals. Beclin1, p62, and LC3B levels were reduced by fructose treatment when taking into account animals born to both CTL and DEX mothers (*p* < 0.05). A closer look through the use of subgroup analysis revealed that beclin1 and p62 levels in DEX-off-FRU, but not in CTL-off-FRU, were lower than those in CTL-off-CTL (approximately 34% lower; *p* = 0.002 and *p* = 0.0006). Besides this, p62 levels in DEX-off-FRU were lower those in CTL-off-FRU (29% lower; *p* = 0.005) (Figure 6A).

## 4. Discussion

Although a consistent body of experimental data reports that fructose intake by rats can lead to hepatic steatosis, hypercholesterolemia, and hypertriglyceridemia [2,3,4], data from human studies suggest that fructose interacts with other environmental variables to modulate energy metabolism [9,10,11,12,13]. The current study addresses this issue by revealing the unprecedented notion that in utero DEX exposure exacerbates fructose-induced hepatic steatosis. We also found that antenatal DEX exposure alone did not sufficiently increase hepatic triglyceride stores in the offspring. As we previously reported, the development of hepatic steatosis in rats born to DEX-treated mothers requires prolonged food deprivation (60 h). Under these conditions, in utero DEX-exposed rats presented a Warburg-like effect in the liver, capable of boosting free fatty acids (FFA) synthesis [18].

Hepatic metabolism of fructose generates intermediates that are subsequently used for glycogen synthesis, gluconeogenesis, or de novo lipogenesis (DNLG) [32,33]. Our data demonstrate that both CTL-off-FRU and DEX-off-FRU exhibit similar levels of hepatic PEPCK activity, as well as pyruvate and glucose intolerance. Previously our group showed that rats exposed to DEX in utero exhibit glucose intolerance and increased gluconeogenesis at 12 weeks of age [18], but herein we demonstrated that these features were absent in the 16-week-old DEX-off-CTL rats. This discrepancy is not surprising, since hepatic glucose production has been shown to progressively increase in rats up until 16 weeks of age [34]. The CTL-off-FRU and DEX-off-FRU groups also exhibited similar glycogen levels, which indicated that the increased hepatic triglycerides stores of DEX-off-FRU animals were not the result of the impaired mobilization of fructose metabolites towards gluconeogenesis or glycogen synthesis.

Despite the fact that we detected reduced hepatic LDH activity and lactate content in the DEX-off-FRU group, it is unlikely that impaired lactate synthesis could be responsible for a relevant deviation of pyruvate to DNLG. In fact, two pieces of evidence support this interpretation. First, pyruvate synthesis itself was probably reduced in DEX-off-FRU, resulting from attenuated PK activity. Second, the activity of CS, the enzyme responsible for converting acetyl-CoA to citrate, was also downregulated in the liver of DEX-off-FRU rats. Hence, our data indicate that funneling fructose intermediates towards DNLG does not have a significant impact on hepatic triglyceride stores in DEX-off-FRU animals.

Since it is known that DNLG also depends on PPP-generated NADPH availability [35], we measured G6PDH activity, the rate-limiting step of PPP, and found that it was equally modulated in the CTL-off-FRU and DEX-off-FRU groups. This result excludes the hypothesis that higher NADPH levels are directed towards DNLG in DEX-off-FRU animals. Additionally, a putative reduction in hepatic fatty acid oxidation in DEX-off-FRU is also unlikely to exacerbate triglyceride stores because the CPT1 activities, a rate-limiting step of fatty acid β-oxidation [36], of the experimental groups were not perturbed when compared to CLT-off-CLT.

Having ruled out the involvement of fructose metabolism as a likely factor for the increased triglyceride stores observed with DEX-off-FRU rats, we then focused on VLDL production as a possible explanation. Indeed, DEX-off-FRU animals displayed lower fasting triglyceride levels and a reduced VLDL production rate.

This observation was further substantiated by downregulated *apob*, *mttp*, and *sec22b* expression in the DEX-off-FRU group. Nascent VLDLs expressing ApoB-100 are synthesized in the ER lumen through an MTTP-dependent process and, before secretion, are transferred to the Golgi apparatus through a process that depends on the snare protein SEC22b [37]. The importance of apob100, MTTP, and SEC22b expression has been previously demonstrated in knockdown and knockout (KO) animal models. For example, siRNA-mediated knockdown of apob100 mRNA was shown to lead to hepatic steatosis [38,39,40], and apob100 knockout (KO) mice develop increased hepatic triglyceride stores [41]. Additionally, MTTP KO mice were also shown to be unable to produce VLDLs, manifesting lower plasma triglyceride levels and hepatic steatosis [42]. Meanwhile, inhibition of SEC22b resulted in impaired triglyceride-enriched VLDL movement from the ER to the Golgi [37].

It is plausible that reduced VLDL production, which would limit the lipoprotein-derived triglyceride supply, could account for the reduced relative weights of the mesenteric and epididymal fat pads of DEX-off-FRU animals. Increases in the relative weights of the mesenteric and epididymal fat pads were only detected in the DEX-off-CTL group. The adiposity of animals born to DEX-treated mothers is still a matter of debate, with reports describing offspring presenting increased, reduced, or unaltered relative weights of the epididymal fat pad, along with reduced body weight [43,44,45]. We believe that the reduced body weight observed in our rats born to DEX-treated mothers results from impaired growth and reduced absolute bone mass. In fact, rats born to DEX-treated mothers have been shown to present reduced femur and tibia lengths and a lower nose-to-anus distance [17].

Besides playing a role in VLDL secretion, SEC22b was also reported to interact with LC3B to participate in autophagy flux [46], a relevant cellular process that delivers lipid droplets to the lysosomes, thus promoting their degradation [47,48]. The inhibition of autophagy has been shown to result in hepatic triglyceride accumulation [49,50]. Autophagic degradation relies on the initial formation of autophagosomes, which occurs through a beclin1-dependent process. LC3B, in turn, mediates the closure of autophagosomes with the sequestration of targets coupled to the polyubiquitin-binding protein p62 [51,52].

In the present study, our results showed that beclin1 levels were reduced in the livers of the DEX-off-FRU rats, indicating that reduced autophagy flux may play a role in the exacerbation of steatosis found in this group. Notably, inhibition of autophagy was also associated with a reduction in VLDL production [53]. Although an abrogation of autophagic flux should lead to an increase in p62 accumulation [54], we found that DEX-off-FRU rats had reduced hepatic p62 content, as well as downregulated beclin1 gene and protein expression. We thus hypothesize that fructose treatment of rats exposed to DEX in utero leads to a reduction in p62 expression. Interestingly, previous studies with p62 KO mice demonstrated that these animals spontaneously develop hepatic steatosis [55,56].

A limitation of our study is the absence of human data correlated to our experimental animal model of intrauterine growth restriction (IUGR). However, considering that there is a considerable amount of data correlating fetal programming with altered postnatal hepatic metabolism reviewed in [57,58], we believe that the relationship between IUGR and fructose consumption should be revisited using previous human data and/or addressed in future trials.

## 5. Conclusions

The present study demonstrated that in utero DEX exposure leads to an exacerbation of hepatic steatosis in adult offspring administered fructose. This unprecedented metabolic programming was not due to intensified funneling of fructose intermediates to DNLG or to impaired fatty acid β-oxidation. Based on our results, it is plausible that impaired VDLD assembly and/or secretion is responsible for the exacerbated hepatic steatosis in DEX-off-FRU rats. This metabolic response also appears to be associated with changes in autophagy proteins.

## Figures and Tables

**Figure 1 nutrients-11-02114-f001:**
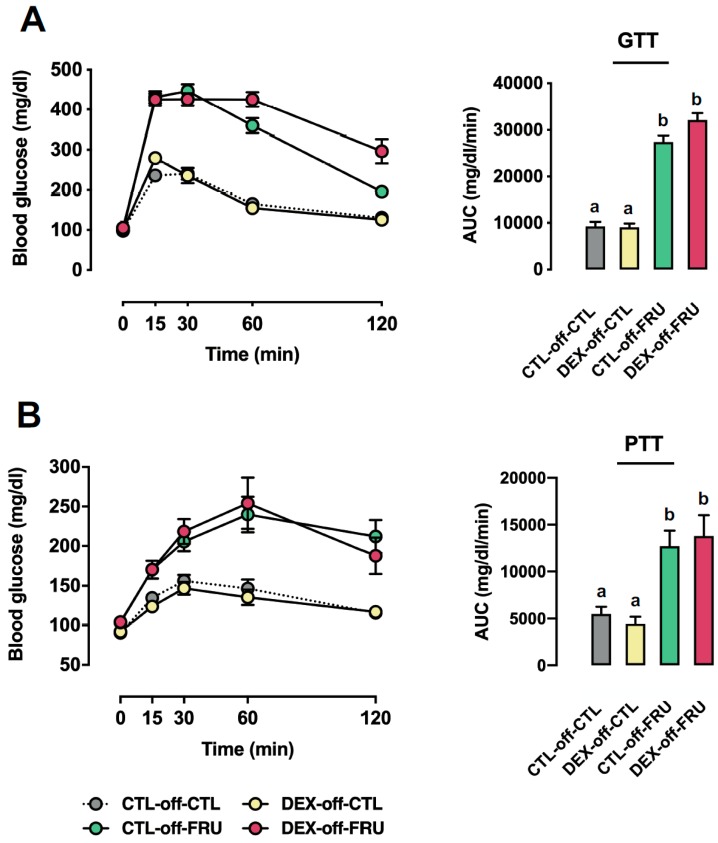
Effects of fructose on glucose homeostasis in rats exposed to dexamethasone (DEX) in utero. CTL-off-CTL, DEX-off-CTL, CTL-off-FRU, and DEX-off-FRU groups were fasted and subjected to either a glucose tolerance test (GTT) (**A**) or pyruvate tolerance test (PTT) (**B**) The areas under the curves were calculated above each individual baseline. The results are presented as the mean ± SE. Means with different superscript minuscule letters are significantly different. The *p* values are provided in Section 3. *N* = 6–8. Area Under the Curve (AUC); Control (CTL); Fructose (FRU).

**Figure 2 nutrients-11-02114-f002:**
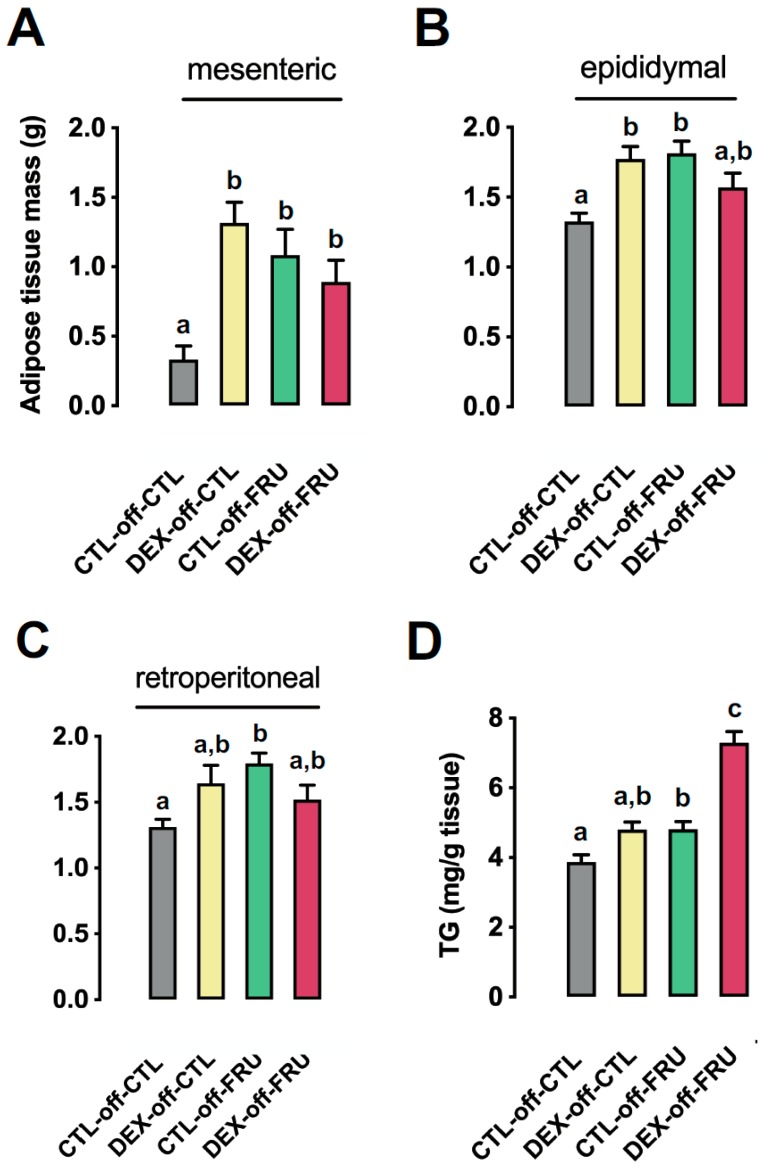
Effects of fructose on the relative fat pad weights and hepatic triglyceride contents in rats exposed to DEX in utero. (**A**) Mesenteric, (**B**) epididymal, and (**C**) retroperitoneal fat pads of the CTL-off-CTL, DEX-off-CTL, CTL-off-FRU, and DEX-off-FRU groups were excised and expressed relative to body weight. (**D**) Fragments of liver were subjected to lipid extraction and subsequent triglyceride determination (**D**). The results are presented as the mean ± SE. Means with different superscript minuscule letters are significantly different. The *p* values are provided in Section 3. *N* = 10–12.

**Figure 3 nutrients-11-02114-f003:**
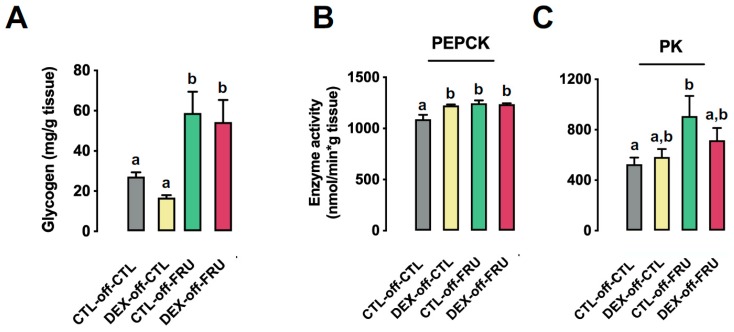
Effects of fructose on the hepatic activity of glycolysis enzymes and levels of glycogen and lactate in rats exposed to DEX in utero. (**A**) Glycogen content of liver fragments from the CTL-off-CTL, DEX-off-CTL, CTL-off-FRU, and DEX-off-FRU groups; (**B**–**D**) phosphoenolpyruvate carboxykinase (PEPCK), pyruvate kinase (PK), and lactate dehydrogenase (LDH) activities, respectively; and (**E**) lactate determinations. The results are presented as the mean ± SE. Means with different superscript minuscule letters are significantly different. The *p* values are provided in Section 3. *N* = 10–12.

**Figure 4 nutrients-11-02114-f004:**
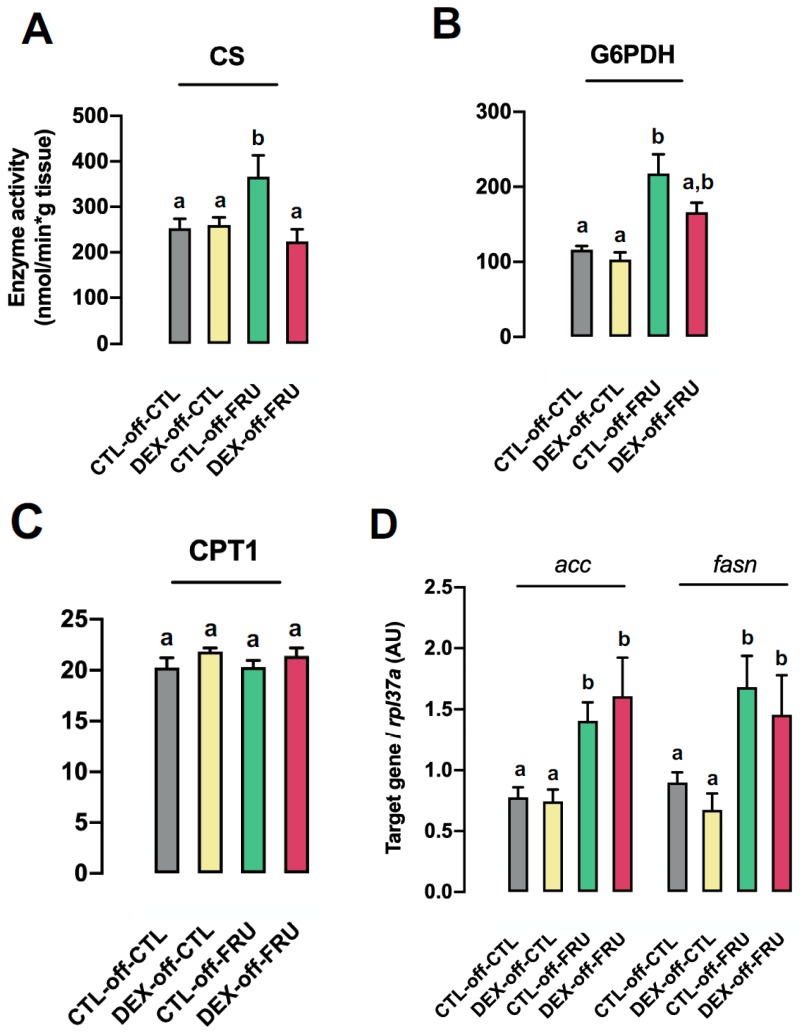
Effects of fructose on hepatic enzyme activities associated with nonesterified fatty acids (NEFA) levels in rats exposed to DEX in utero. The activities of (**A**) citrate synthase (CS), (**B**) glucose 6-phosphate dehydrogenase (G6PDH), and (**C**) carnitine palmitoyltransferase (CPT1) were measured in fragments of liver from the CTL-off-CTL, DEX-off-CTL, CTL-off-FRU, and DEX-off-FRU groups. (**D**) Expression of *fasn* and *acca* in liver fragments, as determined by qPCR. The results are presented as the mean ± SE. Means with different superscript minuscule letters are significantly different. The *p* values are provided in Section 3. *N* = 10–12.

**Figure 5 nutrients-11-02114-f005:**
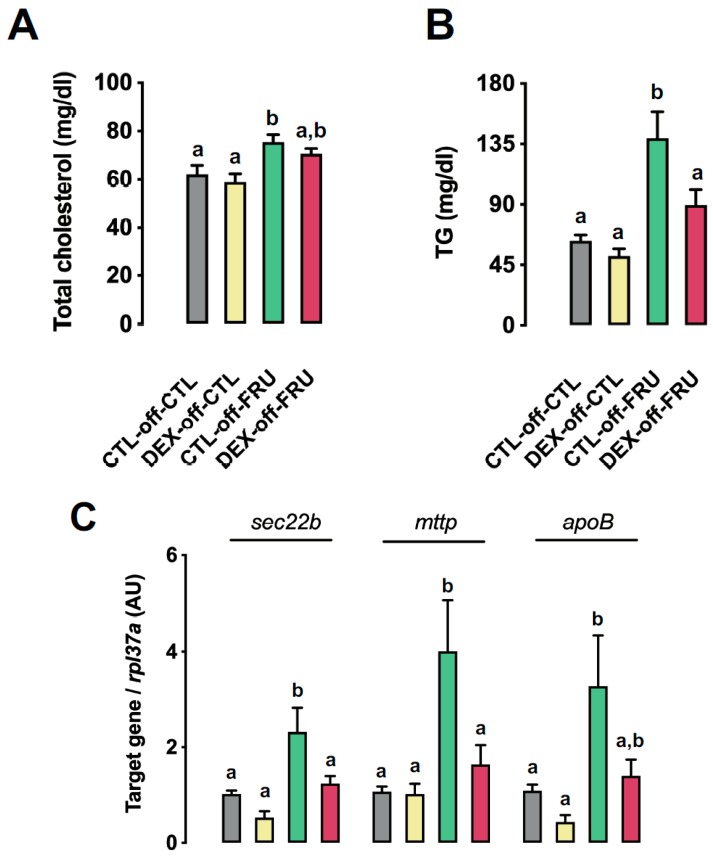
Effects of fructose on parameters related to very-low-density lipoprotein (VLDL) production in rats exposed to DEX in utero. (**A**) Total cholesterol and (**B**) triglyceride (TG) liver content. (**C**) Hepatic gene expression of *sec22b*, *mttp*, and *apoB*, as determined by qPCR. (**D**) VLDL production in CTL-off-CTL, DEX-off-CTL, CTL-off-FRU, and DEX-off-FRU rats injected with tyloxapol. The results are presented as the mean ± SE. Means with different superscript minuscule letters are significantly different. The *p* values are provided in Section 3. *N* = 6–8.

**Figure 6 nutrients-11-02114-f006:**
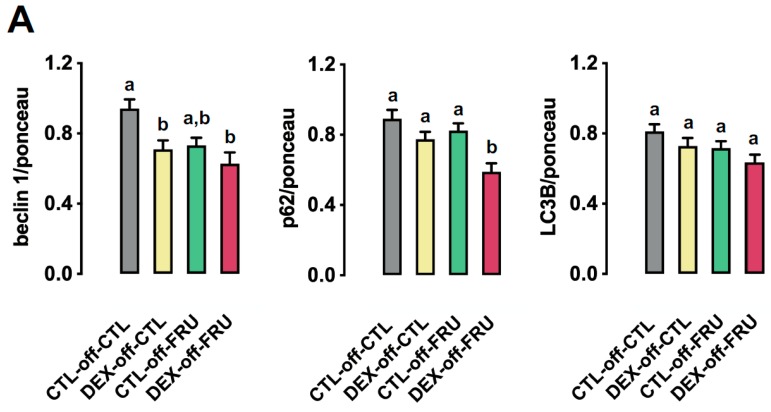
Effects of fructose on autophagy markers in rats exposed to DEX in utero. (**A**) Summary of beclin 1, p62, and LC3B expression in CTL-off-CTL, DEX-off-CTL, CTL-off-FRU, and DEX-off-FRU rats, as determined by Western blots. (**B**) Immunoblots were normalized using Ponceau-stained membranes. The results are presented as the mean ± SE. Means with different superscript minuscule letters are significantly different. The *p* values are provided in Section 3. *N* = 6–8.

**Table 1 nutrients-11-02114-t001:** Nutritional composition of the standard chow (CTL) fed to rats after weaning.

Net protein (%)	22.5
Ether extract (Fat content) (%)	4.5
Carbohydrates (%)	55.0
Fibrous matter (%)	8.0
Ash matter (%)	10.0
Kcal/g	3.5

**Table 2 nutrients-11-02114-t002:** Body weight and caloric intake throughout the experimental period. Means with different superscript minuscule letters are significantly different.

	CTL-off-CTL	DEX-off-CTL	CTL-off-FRU	DEX-off-FRU
Birth weight (g)	7.6 ± 0.2 ^a^(16)	3.7 ± 0.3 ^b^(18)	_	_
Body weight at the 3rd week of life (g)	73.7 ± 6.1 ^a^(16)	39.7 ± 5.2 ^b^(18)	_	_
Body weight at the 8th week of life (g)	350.1 ± 6.8 ^a^(16)	314.2 ± 11.3 ^b^(18)	_	_
Body weight at the 16th week of life (g)	474.6 ± 18.0 ^a^(8)	370.3 ± 10.9 ^b^(7)	510.8 ± 17.2 ^a,c^(8)	439.7 ± 21.3 ^a,b^(11)
Body weight gain during the 1st period (g)	276.4 ± 6.6 ^a^(16)	274.5 ± 13.5 ^a^(18)	_	_
Body weight gain during the 2nd period (g)	119.2 ± 19.4 ^a^(8)	84.7 ± 8.3 ^b^(7)	166.0 ± 10.9 ^a^(8)	107.3 ± 13.7 ^c^(11)
Solid caloric intake during the 2nd period (kcal)	5220 ± 134 ^a^(8)	4863 ± 1123 ^a^ (7)	3884 ± 168 ^b^(8)	4259 ± 28 ^b^(11)
Liquid caloric intake during the 2nd period (Kcal)	_	_	1759 ± 94 ^a^(8)	1350 ± 60 ^b^(11)
Total caloric intake during the 2nd period (Kcal)	5220 ± 134 ^a^(8)	4863 ± 113 ^a^(7)	5642 ± 234 ^b^(8)	5609 ± 293 ^b^(11)

1st period: From weaning to the 8th week of life; 2nd period: from the 9th to the 16th week of life. Means with different superscript minuscule letters are significantly different. The *p* values are provided in Section 3. CTL-off-CTL are the untreated rats born to CTL mothers; CTL-off-FRU are the fructose-treated rats born to CTL mothers; DEX-off-CTL are the untreated rats born to DEX-treated mothers; DEX-off-FRU are the fructose-treated rats born to DEX-treated mothers.

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
