# Peer review of "In Utero Dexamethasone Exposure Exacerbates Hepatic Steatosis in Rats That Consume Fructose During Adulthood"

_nutrients, 2019, doi:10.3390/nu11092114_

Round 1
Reviewer 1 Report
This paper presents data supporting the conclusions that in utero exposure to DEX leads to an exacerbation of hepatic steatosis in adult offspring exposed to fructose. Overall the experiments are well designed and correctly controlled. Just one question arises: what is the exact mechanism through which dexamethasone acts in utero? Dex acts by binding to the glucocorticoid receptor and thereby modulates gene trascription and it can act also through non geniomic pathways. Do the authors have evidences about the type of mechanism?
Author Response
This is a very important issue that we have been working on. As far as we know, DEX directly acts in utero by binding to the glucocorticoid receptor. Unlike endogenous glucocorticoids, DEX is not inactivated by 11-β-hydroxysteroid dehydrogenase 2 in the placenta. Thus, it freely crosses the placental barrier to reach fetal organs. In the fetus, DEX regulates gene transcription; the long-term effects described herein are probably due to the DEX action on the epigenome, including DNA methylation (Crudo et al., Endocrinology 2012, doi: 10.1210/en.2011-2160) and microRNA expression (Pantaleão et al., Sci Rep 2017, doi: 10.1038/s41598-017-10642-1). In the present study we did not attempt to investigate epigenetic mechanisms, so it was not discussed.
Reviewer 2 Report
This is a very interesting study. However, I have some suggestions:
- Please do not refer to “adult life” (see abstract and discussion f.e.) becouse readers might think that results of the study can be perfectly translated to humans and this is not correct. Please, briefly explain in the discussion the differences between animal model and humans. It is not realistic that the steroid expositure can significantly favor (alone) the steatosis of a 60 yrs old man. On the other hand, we know that there are dangerous forms of NASH in childhood and your study can be certainly important specifically for this subgroup. Develop this concept citing one or more papers by Valerio Nobili (Rome) et al..
- Please in the discussion section report limitations of the study.
-Please make a language revision by a mothertongue.
Author Response
- Please do not refer to “adult life” (see abstract and discussion f.e.) becouse readers might think that results of the study can be perfectly translated to humans and this is not correct.
We agree that, as a noun, “adult” must be used for humans. However, as an adjective, the word is applicable for humans and animals.
Please, briefly explain in the discussion the differences between animal model and humans.
Your suggestion was added in the Discussion (including Nobili references). Please see page 18, lines 19-24.
It is not realistic that the steroid expositure can significantly favor (alone) the steatosis of a 60 yrs old man.
We think that our result was misinterpreted. In Figure 2D (TG content in the liver), control and offspring of DEX mothers were not significantly different.
However, extensive epidemiological studies have reported associations between characteristics of early development and health outcomes later in life. The exposure of the fetus to increased levels of glucocorticoids, resulting from maternal stress or treatment with synthetic glucocorticoids, can lead to long-term 'programming' with cardiovascular and metabolic consequences. The best example is the peri-conceptional exposure to famine during the Dutch Hunger Winter in 1943–1945.
On the other hand, we know that there are dangerous forms of NASH in childhood and your study can be certainly important specifically for this subgroup. Develop this concept citing one or more papers by Valerio Nobili (Rome) et al.
We started the fructose administration in adult rats (2 months of age), so we considered that including a discussion of NASH in childhood would take the focus away from our results.
- Please in the discussion section report limitations of the study.
Thank you for your suggestion. It was added to the Discussion. Please see page 18, lines 19-24.
-Please make a language revision by a mothertongue.
This has been done.
Reviewer 3 Report
Payolla et al. investigate the role of in-utero dexamethasone (DEX) exposure to the metabolic response to high-fructose diet. Given the widespread use of glucocorticoids in pregnant women and the prevalence of fructose in the diet, this is a medically important area of investigation. The major finding of the present study is that in utero DEX treatment exacerbates fructose-induced hepatic steatosis in adult rats. The authors also establish reduced VLDL secretion as the mechanism underlying hepatic TG accumulation.
While the impact of prenatal DEX exposure on hepatic steatosis has been investigated in multiple previous studies, the observation on fructose-induced fatty liver in the present work is novel. However, the manuscript has several major weaknesses as follows:
1. In a previous publication (ref. 18), the authors reported major effects of DEX exposure on glucose and pyruvate tolerance after 12 hours of fasting. These results contrast with results presented in the current study, i.e. no difference between CTL-off-CTL and DEX-off-CTL (Fig. 1). Similarly, the authors have not been able to recapitulate the previously observed effect of DEX on steatosis (Fig. 2D). The source of these discrepancies is unclear and it is not discussed in the manuscript.
2. A lack of absolute body weight (BW) data is a major deficiency of this manuscript. As only weight gains are reported, the BW of experimental animals is unclear. This is a critical issue, as BWs do seem to be different in experimental groups. For example, DEX treatment reduces birth weight as well as weight gain during the 2nd period. This means that DEX-off rats may weigh 30-60 g less than their CTL-off counterparts. This information would be critically important to the interpretation of metabolic analyses. Absolute BW at weaning, 8-w and 16-w should be reported.
3. From data presented in Figure 2, it seems that DEX treatment increases adiposity in CTL mice (i.e. CTL-off-CTL vs DEX-off-CTL). How can this be reconciled with reduced weight gain in DEX-off-CTL?
4. At multiple occasions, the authors draw conclusions that are not supported by the data presented:
Line 222-223: “… the activity of PEPCK…was equally increased by treatment with fructose irrespective of previous exposure by DEX”. Data in Fig. 3B are inconsistent with this statement (i.e. DEX-off-CTL is not different from DEX-off-FRU).
Line 243-244: “… the activity of G6PDH…was increased by fructose treatment irrespective of previous exposure to DEX”. Data in Fig. 4B are inconsistent with this statement (i.e. DEX-off-CTL is not different from DEX-off-FRU).
Line 284-285: “… beclin1…were reduced by fructose treatment irrespective of previous exposure to DEX”. Data in Fig. 6A are inconsistent with this statement (i.e. DEX-off-CTL is not different from DEX-off-FRU).
5. Normalization of enzymatic activities by liver weight is inappropriate considering the differences in glycogen/TG content. Total protein should be used for normalization. It is also unclear how protein expression was “normalized by Ponceau-stained membranes” (Fig. 6)
6. The analysis of TG secretion data is inconsistent with section 2.8. in the methods section. It is stated that secretion rate was calculated by linear regression, whereas Fig. 5D shows that it was assessed by AUC analysis, which is inappropriate. Furthermore, the non-linear increase of serum TG (even reduction at 1 h!) suggests technical problems. The time-points listed in section 2.8 do not reflect this experiment. Hours are incorrectly labeled as minutes.
7. The authors exclude the role of de novo lipogenesis (DNLG) based on measurements of glycolisis/TCA/PPP enzymes. This claim remains completely unsupported in the absence of assessing enzymes involved in DNLG (Acc1, Acly, Fas, Me1 etc.).
8. The manuscript contains numerous typos and grammatical errors. It is suggested that the text is revised by someone with proficiency in English writing.
Author Response
In a previous publication (ref. 18), the authors reported major effects of DEX exposure on glucose and pyruvate tolerance after 12 hours of fasting. These results contrast with results presented in the current study, i.e. no difference between CTL-off-CTL and DEX-off-CTL (Fig. 1). Similarly, the authors have not been able to recapitulate the previously observed effect of DEX on steatosis (Fig. 2D). The source of these discrepancies is unclear and it is not discussed in the manuscript.
This is a very pertinent observation. Our previous publication reported glucose intolerance and increased gluconeogenesis in 12-week old rats born to DEX-treated mothers (Pantaleão et al., 2017). The current study, instead, evaluated glucose and pyruvate tolerance in 16-week old rats born to DEX-treated mothers. The present data revealed that, with this age, rats born to DEX-treated mothers do not manifest glucose intolerance and/or increased whole-body gluconeogenesis. Our understanding is that the continuous and age-related increase in hepatic glucose production that occurs in rats until 16 weeks of age (Barzilai & Rossetti, Age-related changes in body composition are associated with hepatic insulin resistance in conscious rats – 1996) may have masked pyruvate and glucose intolerance induced by prenatal exposure to DEX in the present experiments. We have added these interpretations in the revised manuscript. Please, see page15, lines 23-25, and page 16, lines 1-2.
Regarding the effects of antenatal DEX of hepatic TG content, our previous publication reported steatosis in 60h-fasted rats born to DEX-treated mothers (Pantaleao et al., 2017). All the samples obtained in the present study were from 12h-fasted rats. As discussed in our previous publication, prolonged fasting (60 hours) elicits a Warburg-like effect in the liver of rats born to DEX-treated mothers, shunting pyruvate and other intermediates to FFA synthesis. Although it is not the scope of the present study to reveal the mechanism underlying such differences, 12h-fasted rats born to DEX-treated mothers only developed hepatic steatosis following fructose treatment. The experimental differences between our previous publication and the current experiments are now acknowledged in the revised version of the menauscript. Please see page 15, lines 13-18.
A lack of absolute body weight (BW) data is a major deficiency of this manuscript. As only weight gains are reported, the BW of experimental animals is unclear. This is a critical issue, as BWs do seem to be different in experimental groups. For example, DEX treatment reduces birth weight as well as weight gain during the 2ndperiod. This means that DEX-off rats may weigh 30-60 g less than their CTL-off counterparts. This information would be critically important to the interpretation of metabolic analyses. Absolute BW at weaning, 8-w and 16-w should be reported.
We have added the absolute body weights at the 3rd, 8thand 16thweeks of age (please, see new Table 2). As it can be noted, body weights of rats born to DEX-treated mothers are lower than those of the rats born to CTL mothers at these three time-points. These reductions do not impact body weight gain during the first period (between the 3rdto 8thweeks) but does result in a lower body weight gain during the second period (between the 8thand 16thweeks). These results have been included in the revised manuscript. Please see page 9, lines 15-25, and page 10, lines 1-2.
From data presented in Figure 2, it seems that DEX treatment increases adiposity in CTL mice (i.e. CTL-off-CTL vs DEX-off-CTL). How can this be reconciled with reduced weight gain in DEX-off-CTL?
We did find an increase in relative weight of mesenteric and epididymal fat pads of DEX-off-CTL (when compared to CTL-off-CTL). On the other hand, as also indicated in the original version of the manuscript, the relative weight of the retroperitoneal fat pad was not altered in DEX-off-CTL. Hence, we cannot attest that in the present study rats born to DEX mothers presented increased whole body adiposity (although a tendency towards an increase would be a likely result). Adiposity in the offspring born to DEX-treated mothers is still a matter of debate. Published papers described that rats born to DEX-treated mothers present an increase or decrease in relative weight of the epididymal fat pads in parallel to a reduction in body weight (Sugden et al., Maternal glucocorticoid treatment modulates placental leptin and leptin receptor expression and materno-fetal leptin physiology during late pregnancy, and elicits hypertension associated with hyperleptinaemia in the early-growth-retarded adult offspring - 2001, Eur J Endocrinol; Zulkafli et al., Postnatal Dietary Omega-3 Fatty Acid Supplementation Rescues Glucocorticoid-Programmed Adiposity, Hypertension, and Hyperlipidemia in Male Rat Offspring Raised on a High-Fat Diet - Endocrinology, 2013). Still, others have described that the relative weight of epididymal fat pads remained unchanged in the offspring born to DEX-treated mothers (Wyrwoll et al., Prevention of Programmed Hyperleptinemia and Hypertension by Postnatal Dietary omega-3 Fatty Acids – Endocrinology, 2006).
Our interpretation is that the reduced body weight presently described in the DEX-off-CTL group might rather result from impaired growth and reduced absolute bone mass. Corroborating this hypothesis, rats born to DEX-treated mothers have been described to present reduced femur and tibia length and lower nasal-to-anus distance (Carbone et al., Prenatal Dexamethasone Exposure Potentiates Diet-Induced Hepatosteatosis and Decreases Plasma IGF-I in a Sex-Specific Fashion – Endocrinology, 2012). We have now considered these aspects in the Discussion of the revised manuscript. Please, see page 17, lines 17-25.
At multiple occasions, the authors draw conclusions that are not supported by the data presented:
Line 222-223: “… the activity of PEPCK…was equally increased by treatment with fructose irrespective of previous exposure by DEX”. Data in Fig. 3B are inconsistent with this statement (i.e. DEX-off-CTL is not different from DEX-off-FRU).
Line 243-244: “… the activity of G6PDH…was increased by fructose treatment irrespective of previous exposure to DEX”. Data in Fig. 4B are inconsistent with this statement (i.e. DEX-off-CTL is not different from DEX-off-FRU).
Line 284-285: “… beclin1…were reduced by fructose treatment irrespective of previous exposure to DEX”. Data in Fig. 6A are inconsistent with this statement (i.e. DEX-off-CTL is not different from DEX-off-FRU).
When performing the two-way ANOVA, our analysis indicated that the factor “treatment with fructose” increased PEPCK and G6PDH activities and reduced beclin1 levels. Meaning that these differences occurred when comparing all rats receiving fructose (both the CTL and the DEX offspring) to all those receiving tap water (both the CTL and the DEX offspring). We agree with the reviewer that our original description did not make this more clear. We have made changes throughout the text in order address this concern.
With regards to the sub-group comparisons, it is important to emphasize the following aspects:
1: Indeed PEPCK activity of DEX-off-CTL is not different from DEX-off-FRU. However, this activity is significantly elevated in the CTL-off-FRU and DEX-off-FRU groups, when compared to CTL-off-CTL (14% and 13%; P=0.002 and P=0.003, respectively).
2: G6PDH activity in DEX-off-FRU is 61% higher than that of DEX-off-CTL (P=0.03). G6PDH activity in CTL-off-FRU is also 87% higher than that of CTL-off-CTL (P<0.0001). Nevertheless, G6PDH activity in DEX-off-FRU is similar to CTL-off-FRU.
We have changed the text to clarify the difference mentioned make the above. As we originally interpreted, these results still corroborate our original conclusion that DEX-off-FRU and CTL-off-FRU do not have different PEPCK and G6PDH activities that could account for the increased hepatic TG levels.
3: Indeed, the beclin1 levels of DEX-off-CTL are not different from those of DEX-off-FRU. However, when compared to CTL-off-CTL, beclin1 levels in the DEX-off-FRU were reduced by 35% (P=0.002). A result that was not observed with CTL-off-FRU. Such data support our additional interpretation that a reduction in autophagic flux occurs only in DEX-off-FRU rats. Such a modulation may explain the increased steatosis seen in this experimental group.
Normalization of enzymatic activities by liver weight is inappropriate considering the differences in glycogen/TG content. Total protein should be used for normalization. It is also unclear how protein expression was “normalized by Ponceau-stained membranes” (Fig. 6).
Thank you for your observation. However, we have a different understanding about this topic. We believe that the biological relevance of the enzymatic activity data is better evidenced by normalizing it by a given mass of the sample. In other words, even if the changes in the activity of a specific enzyme in a fragment of liver is a result of changes in its whole protein composition, it would still exert biological impacts.
Normalization of protein expression was performed as follows: Nitrocellulose membranes were stained with ponceau before incubation with antibodies. Stained membranes were scanned and digital images were acquired. Immunoreactive bands were detected using chemiluminescent reaction and visualized with an Amersham Imager 600 device. Densitometric analyses of ponceau stained membranes was reported to be as reliable as detecting actin levels by means of stripping and reprobing the same membrane (Romero-Calvo et al., Reversible Ponceau staining as a loading control alternative to actin in Western blots – 2010). We used this normalization procedure in two recent studies (Teixeira et al., Life Sciences – 2018; Vicente et al., Life Sciences – 2019). We have changed the revised manuscript in order to provide a more detailed description of this normalization procedure. Please see page 7, line 25, and page 8, lines 1-3.
The analysis of TG secretion data is inconsistent with section 2.8. in the methods section. It is stated that secretion rate was calculated by linear regression, whereas Fig. 5D shows that it was assessed by AUC analysis, which is inappropriate. Furthermore, the non-linear increase of serum TG (even reduction at 1 h!) suggests technical problems. The time-points listed in section 2.8 do not reflect this experiment. Hours are incorrectly labeled as minutes.
The units of the time points and the y-axis label of Fig 5D were wrong. Thank you for your observation. The revised Figure 5D is now correctly labeled. No technical problems occurred during the VLDL production assay. The unaltered TG levels 1 hour after tyloxapol injection (and even a slight non-significant trend towards reduction) comprises the typical changes in TG levels when the test is performed using intraperitoneal injections. In contrast to the intravenous tyloxapol injection, intraperitoneal tyloxapol injection does not lead to an immediate increase in blood TG levels. These differences in TG dynamics occur because the high molecular weight of tyloxapol which may delay its absorption from intraperitoneal space.
The reviewer may find concordant changes in blood TG levels in papers published by other groups using intraperitoneal tyloxapol injection (Mulvihill et al., Naringenin Prevents Dyslipidemia, Apolipoprotein B Overproduction, and Hyperinsulinemia in LDL Receptor–Null Mice With Diet-Induced Insulin Resistance - Diabetes, 2009; Li et al., Transgenic expression of cholesterol 7alpha-hydroxylase in the liver prevents high-fat diet-induced obesity and insulin resistance in mice – Hepatology, 2010; Issandou et al., Pharmacological inhibition of Stearoyl-CoA Desaturase 1 improves insulin sensitivity in insulin-resistant rat models – Eur J Pharmacol, 2009; Vujic et al., Monoglyceride lipase deficiency affects hepatic cholesterol metabolism and lipid-dependent gut transit in ApoE−/− mice – Oncotarget, 2017).
The authors exclude the role of de novo lipogenesis (DNLG) based on measurements of glycolisis/TCA/PPP enzymes. This claim remains completely unsupported in the absence of assessing enzymes involved in DNLG (Acc1, Acly, Fas, Me1 etc.).
We agree with the reviewer. We have now assessed the mRNA expression of fasn and acca. We found that the factor “exposure to fructose” increased the expression levels of fasn and acca, when taking into account animals born to both CTL and DEX mothers (69% and 97% higher than animals not treated with fructose; P=0.04 and P<0.003, respectively). However, the subgroup analysis revealed no differences between CTL-off-FRU and DEX-off-FRU. These data corroborated our initial proposition that differential modulation of DNLG cannot explain the differences between the hepatic TG levels of CTL-off-FRU to DEX-off-FRU rats.
These new data are now included as figure 4D.
The manuscript contains numerous typos and grammatical errors. It is suggested that the text is revised by someone with proficiency in English writing.
We apologize for the typos and grammatical errors. A native English speaker has reviewed the new version of the manuscript.
Round 2
Reviewer 2 Report
The modified version of the study is printable.
Reviewer 3 Report
Reviewer's comments have been adequately addressed.